# Perceiving and Adapting to Climate Change: Perspectives of Tuscan Wine-Producing Agritourism Owners

**Rachel Germanier ***  **and Niccolò Moricciani**

Les Roches Global Hospitality Education, CH-3975 Crans-Montana, Switzerland
* Correspondence: rachel.germanier@lesroches.edu

**Abstract:** It is now widely accepted that climate change is having a profound impact on the weather systems around the world. These, in turn, have a considerable effect on two important elements of the Tuscan economy: wine production and tourism. This case study sought to explore the relationship between the perception of Tuscan wine-producing agritourism owners of the potentially abstract notion of climate change and their concrete experiences as entrepreneurs. Semi-structured interviews were conducted with eight wine-producing agritourism owners or managers in Val d'Orcia, a small area of Siena, Tuscany, and analysed thematically. The impact of climate change on the area's viticulture is undeniable but the responses to the challenges are more nuanced. Political leadership on the climate crisis appears absent and perhaps as a consequence, these small-scale operators lack knowledge and funds to enable them to plan ahead: they react often day-to-day to the immediate weather conditions rather than planning long term. While recognizing the difficulties they face from climate change as viticulturists, as agrotourism owners they welcome the longer seasons which enable them to open in the formerly barren shoulder seasons but struggle with last-minute cancellations due to unpredictable weather in the area.

**Keywords:** Tuscany; Val d'Orcia; climate change; viticulture; viniculture; agritourism

## 1. Objective

Climate change, "long term shifts in temperatures and weather patterns" [1], is considered as one of the most important problems of our time [2]. It creates various threats, not only to human health, but it also causes damage for agriculture through extreme weather conditions, water scarcity, heatwaves and insect migration [3]. Wine is one of the most popular beverages across the globe and is produced only in specific regions around the world [4]. Winemaking is strongly connected with rural tourism, and many countries identified enotourism as a highly beneficial development strategy [5]. This paper will initially explore the literature related to climate change: both the realities and individuals' perception of it, then examine the ways climate change is impacting viticulture and tourism in rural areas with a focus on Italy and the Val d'Orcia in particular and on how small-scale wine-producing agritourism owners are adapting to these changes.

### 1.1. Climate Change—Perception, Reality and Adaptation in Viticulture

It is now generally accepted that climate change has become a climate crisis [6] and climate change is affecting weather systems globally [7]. The currently known effects include higher temperatures, more frequent wildfires and droughts, changes to precipitation patterns and an increase in density of storms [8]. Despite there now being a scientific consensus that the planet is warming [9], Fourment et al.'s [10] recent work exploring wine growers' perception of climate change still found scepticism on this topic. Notwithstanding this example of an unwillingness to face the truth, there is a broader swathe of research, dating back to at least 2005 [11] and up to the present day [12], exploring how climate change is affecting wine growing and its interrelated tourism. Climate change has been

demonstrated to have an impact worldwide on winemakers [11,13,14], from Canada [15] to South Africa [16] and on into Europe [17,18], and indeed on wine tourism [19,20], despite a heavy research emphasis on wine tourism/production and sustainability - see for example [21,22]. Droulia and Charalampopoulos [23] stand out by providing an excellent overview of the state of the academic knowledge regarding climatic impacts on European viticulture as of 2021.

Research in China [24] has shown that experiencing extreme weather events is correlated with a belief in climate change while slightly older work from America [25] has demonstrated that the weather at the time of the questioning can affect respondents' answers regarding climate change. Moving to Europe, a survey conducted in 2016 looking at Norwegian, British, French and German perceptions of climate change [26] found that "while most people in all four countries are to some extent worried about climate change, very few express high levels of worry" (p. 36). The social perception of climate change has been explored in the Italian context [27], but not specifically in the field of agriculture or viticulture. Gioia et al. [28] also investigated the context of Italy, specifically the north, and similarly found an understanding among citizens of the threat of climate change, more specifically geo hazards, but again this was not specifically among people working with the earth. In 2009, Battaglini et al. explored German, French and Italian winegrowers' perception of climate change and concluded that winegrowers in Italy (with a high percentage from Tuscany) were cognizant of specific elements of climate change which were affecting their local weather systems: more irregular weather patterns, higher summer temperatures, less precipitation, late frosts and sometimes wet weather causing an increase in pests and diseases. Powdery mildew and hailstorms have also been identified as being problematic by Italian farmers [29]. Several studies have focused on vulnerable Italian wine growing areas to climate variation [30–34]. A relative early work in the field was by Spellman [35] who laid out the importance of terroir and described how weather events such as rain, hail, wind, etc. can impact viticulture. Other works have linked these weather events to a pattern of climate change [36,37]. Exploring the Tuscan context specifically, it was demonstrated in 2010 [38] that there has been decreased rainfall and higher temperatures leading to an increase in the area capable of grapevine cultivation, a reduced yield and potentially shifting grapevine cultivation to higher elevations. It is not known to what extent these problems are still relevant nearly 15 years later in the Tuscany area, bearing in mind that the planet has continued to warm during this period [9], and the region experienced major wildfires in 2021 [39].

The challenges continue for the vintner in the cellar. According to Mira de Orduña [37], they include "higher temperatures of harvested grapes delivered to the winery, higher environmental temperatures during fermentations, higher grape berry sugar and, possibly, potassium concentrations, lower acidity levels and higher pH values" (p. 1848).

Vintners, faced with these meteorological challenges, are forced to adapt their practices, both in the vineyard and in the cellar. The potential to adapt seems dependent on the scale of the operation with a study indicating that the Canadian industry's smaller players are likely to be less adapted to events [15] than their larger counterparts. While wine growers are vulnerable to climate change, major adaptations such as moving vineyard location and changing vine varieties have the largest impact, but are not as readily implemented as grape growing and wine making processes [40]; indeed it has been demonstrated that the ability to adapt to climate change in Europe is limited [41]. Strategies that have been shown to be implemented globally include: irrigation, tilling strategies and aligning vines [36], changing the pruning time [42], growing a cover crop, leaving more leaves on the vines, and harvesting earlier [37,41]. Quénol et al. [43] focuses on the French situation and quotes Neethling et al. (2013) dividing the adaptations temporally into short term: harvest management, winemaking, canopy management and soil management; medium term: changing rootstock, site selection locally and row orientation, and long term: introducing new grape varieties and irrigation. Holland and Smit [44] acknowledge that adaptive strategies are learned through trial and error and are reactive in the short term.

Neethling et al. [45] recognise the uncertainty faced by winegrowers which underpins this short-term vision, while Bernetti et al. [19] express the importance of the identity of the geographical area and how farmers respond culturally to these changes which can potentially upset so many of the ideas they hold dear. The literature provides many adaptive solutions which can be implemented reactively and highlights others which require a more long-term and financially important investment which it seems likely small-scale operators do not possess. Indeed, it has been pointed out that winegrowers' real experiences are rarely explored in the literature [40] and that there is a lack of knowledge about which strategies for adapting to climate change in Italy are relevant [31], so an investigation into small-scale winegrowers' practices seems timely to see how they manage, successfully or otherwise, the conundrum of adapting with limited financial means.

While considerable academic energy has been spent on describing the effects of climate change and possible adaptive solutions for viticulturists, there has been a narrower exploration of adaptive measures in the wine cellar. Mozell and Thach [14] summarised the literature available in 2014 and produced a comprehensive 25-step guide for mitigating and adapting to climate change in both the vineyard and the winery, which are returned to in Section 3.2.2 below. Having explored the literature relating to climate change and winegrowing, the next section will address how the academic world perceives the impact of climate change on a specific form of rural tourism, agritourism.

### 1.2. Climate Change—Perception, Reality and Adaptation for Agritourism

It is necessary firstly to define agritourism. It is challenging to identify one single definition of the term as there is, according to Roman et al. [46], disagreement about the definition in the literature. A pertinent definition for this work on how Italians perceive the term can be ascertained by summarizing the Italian website for Agritourisms' introduction [47]. From there, one can determine that: "Agritourism's unique feature is that it can only be practiced on farms and by farmers. It is a cultural phenomenon with the farm providing a complete but simple rural experience". This definition includes the term "farm" which Philip et al. [48] included in their typology and stresses the rural experience for the tourist which appears significant in the Italian context as it is mentioned twice on the Agriturismo Italia website home page [47]. It is also highly relevant for the context of this study, based as it is on rural, small-scale farms which often emphasize their small, family-run nature. Located in often scenic locations, agritourisms offer tourists an opportunity to experience nature [49], engage in experiential dimensions [50] and "taste the local" [51]. As such, they need to remain true to the traditions of the locale yet manage the challenges climate change is throwing at them. No literature could be found specific to perception of Italian tourism agents with respect to climate change. The realities of these changes will be explored next.

The United Nations, at Cop26 [52], stated that hotter temperatures, more severe storms, and increased drought are among the eight effects of climate change globally. Studies have been carried out exploring the effects of this climate change on rural and mountain tourism in Europe. These have demonstrated that what are described as 'shoulder seasons', i.e., spring and autumn, either have already, or will be, perceived as desirable periods for vacations [12,53–55] and enable wineries to exploit their wine making and tasting facilities in the seasons when they are otherwise underused [12].

The excessive heat in the peak season in Tuscany specifically is described as potentially resulting in a significant decline in the number of domestic tourists visiting the area [56], but the same study indicated that international tourists seem more resilient to these temperatures. Studies on wine tourism demonstrate that the clement weather in these inevitably rural areas is found to be attractive to customers [57], but it is not known if this also applies to visitors to agritourisms.

Regarding adaptation, the literature sees agritourism as a means for farmers globally to supplement their income in a sustainable manner (see [58–60]), but no research has been

identified which explores how agritourism in Tuscany can adapt to the effects of climate change they are facing, apart from opening in the shoulder seasons mentioned above.

### 1.3. Research Questions which Emanate from the Literature Review

Three distinct research questions arose from this literature review and were addressed individually, exploring responses in viticulture and tourism separately. They will be addressed in the following sections:

RQ1: How do Tuscan wine-producing agritourism owners perceive climate change?

RQ2: How are Tuscan wine-producing agritourism owners affected by and how do they adapt to climate change?

RQ3: How does climate change affect the agritourism business for Tuscan wine-producing agritourism owners and how do they adapt to these impacts?

## 2. Material and Methods

### 2.1. The Study Area

The design of this research was a typical case study, exploring the "everyday or commonplace situation" [61] in a "bounded" place [62], where the location, in this case the Val d'Orcia area in the Siena province of Tuscany (see Figures 1 and 2), was important [63].

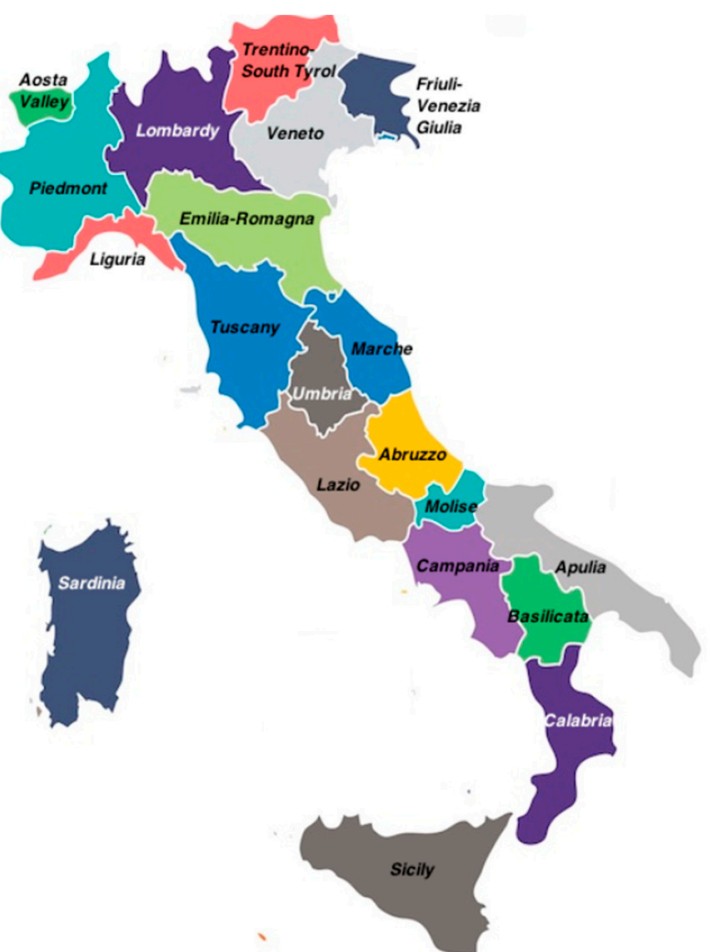

**Figure 1.** Map of the regions of Italy Reprinted with permission from Touropia.com. 2022, Touropia.com [64].

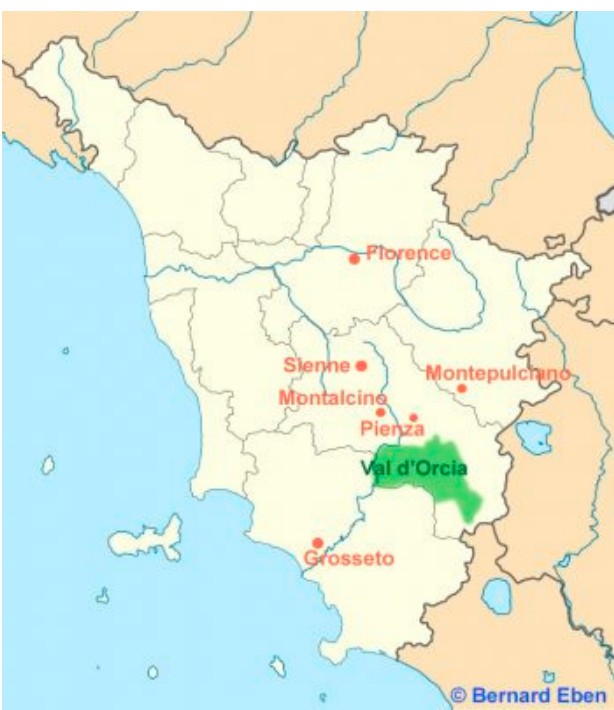

**Figure 2.** Map of Tuscany with the small area of Val d'Orcia in the south east coloured in green [65]. Reprinted with permission from Bernard Eben. 2017, Bernard Eben.

This picturesque region is a UNESCO world heritage site whose wines and landscapes are part of its attraction for tourists [66,67], (see [68]). According to Agriturismo.it [47], there are 1172 forms of agritourism in the Tuscany area, and 85 in Val d'Orcia, demonstrating the popularity of this form of holiday accommodation in the region. Young [69] explains that the wineries in the area tend to be family run, organic, and focused on small-scale production with "high" wine quality. Many of the vintners also cultivate olives and cereals and, as typified by The Store of Valdorcia [68], have a strong attachment to tradition as well as the land they farm. The agriculture and the tourism element are therefore interdependent.

### 2.2. Data Collection and Analysis

The data collected were qualitative in nature, coming from semi-structured interviews with eight purposefully selected wine-producing agritourism owners/managers in the Val d'Orcia region in Tuscany in the spring of 2021. The owners were "typical" [70] of those in the area, and yet an under-researched group. They were known to the researchers; indeed, we deliberately chose the position of interviewer as an insider in order to benefit from the "trust and openness" [70] that insider research can bring. Each interview was undertaken in Italian, then recorded and translated for analysis by the interviewer into English. The interview schedules (Appendix A in English and Appendix B in Italian) were designed to gather data which fed into the three themes of the RQs mentioned in 1.3. The semi-structured approach provided reliability and comparative data [71] while giving the interviewer scope to engage with insider-informed comments, thus reinforcing rapport with the interviewees [62]. The transcripts were imported into MAXQDA for a hybrid deductive/inductive content/thematic analysis. Each interview was coded for deductive codes (for example, codes relating to climate-change impacts on the vines, and also for inductive codes when content or themes not mentioned in the literature were alluded to by participants, such as their perceptions of climate change and their reactions to climate change). The overarching themes were broken down into more specific codes which were used on each interview transcript. By working deductively and inductively to establish patterns, by giving voice to the respondents and by trying "to understand their setting", the work is typically qualitative, as Creswell [62] outlines.

We ensured internal validity, i.e., ensuring that the findings were sound [70] and could be trusted [72] by first undertaking a pilot study, in Italian, with a suitable candidate whose knowledge about both winemaking and agritourisms meant his contributions were deemed valid. Following this pilot, two questions were modified to make them clearer to the interviewees. Regarding external validity, as the sample was only of eight individuals, there was no intention to generalize to a wider population or to attempt to replicate our findings. Ethical clearance for undertaking this work was obtained from the researchers' institution and informed consent was sought and obtained from all participants before the interviews began.

## 3. Results and Discussion

*3.1. How do Tuscan Wine-Producing Agritourism Owners Perceive Climate Change?*

As individuals working "and living every day in the countryside", the smallholders are acutely aware of the changes to the weather systems that affect them on a daily basis. This was epitomized by R8 who commented: "I . . . feel its effect every day on all my cultures such as grapes, cereals and olive oil" and who said he realised that the cause of these weather irregularities is climate change. There is a perception that climate change is indeed "tangible" (R1) and "affecting our cultures and the way we are working" (R6), echoing the findings of Dai et al. [24]. There is a sense of foreboding and worry that the changes witnessed are out of their control and that they are losing their habitual synergy with nature; indeed, R7 announced that "for sure, the seasons went crazy". R2 summarised the overarching sentiment of the farmers: "global warming at the moment [is] really worrying us".

While some respondents referred to the impacts of climate change in the future tense, their accounts of the way their work on the farm has been and is being impacted by the "crazy seasons" is completely in line with the current academic literature. This recognition of the immediate impacts of climate change is to be contrasted with their more reactive and sometimes passive attitude to how to adapt in the long term which will be discussed below.

*3.2. How Are Tuscan Wine-Producing Agritourism Owners Affected by and How do They Adapt to Climate Change?*

All the agritourism owners in our study were also wine producers, although the relative economic importance of each enterprise differed among the interviewees. However, as agritourisms are, by definition, agricultural enterprises, the tourism element is dependent on there being a profitable agricultural business, so their understanding of and adaptation to climate change are vital to comprehend.

This RQ is divided into firstly the effects, and then the adaptations for ease of comprehension for both sectors of the smallholders' enterprises.

### 3.2.1. The Effects of Climate Change on Grape Growing
The Effects of the Weather

The rise in temperatures [8,9], increased drought and changes to precipitation patterns [8] have all been noticed by the farmers and attributed to climate change in accordance with the UN's findings [52]. The summers are becoming "extremely hot" (R7), and humidity is rising too (R3), but also the winters are warm when, as the farmer who is typically in tune with his land states, "it is supposed to be cooler" (R8). These extremes, including warm days and cold nights, "passing from summer to winter in one night" (R3), "can create serious damage" (R7). Hot temperatures are not the only problem the wine producers face. During the spring of 2021, when we were gathering our data for this project, severe and unusually late frosts hit Tuscany, leading to farmers trying to protect "the grapes that had already developed due to the heat of January and February" (R4). According to our respondents, these problematic late frosts are becoming more common. The excessive heat in summer is accompanied by "long periods without water and then there is enormous

precipitation and in one day as much water falls as normally in four months" (R3). Indeed, R5 explains that "we witness waterbombs which destroy everything".

The hotter weather is accompanied by periods of drought. Overall, there is less precipitation than before—notwithstanding the aforementioned waterbombs—and water is described as "lacking" (R2) and "missing" (R4)—reinforcing the notion of deviation from what is natural, normal and expected.

Supporting the findings of Menapace et al. [29], more frequent hailstorms (R1) which can ruin entire harvests by damaging the grape berries are considered to be "one of the biggest problems" (R3). As R1 explained: "I cannot remember a summer without a hailstorm since 2014" but on a more positive note, they appear very localised and although they are "becoming stronger" (R8) an individual farm might be spared "extreme damage" (R3).

The findings of Battaglini et al.'s [17] sensitive work with farmers in Europe aligned closely to the data we collected. The excess heat, the rising humidity, the unpredictable nature of the weather systems - more irregular weather patterns, higher summer temperatures, less precipitation, late frosts and sometimes wet weather causing an increase in pests and diseases which they mentioned 13 years ago - are still valid today, reinforcing the notion that the weather patterns from before are not going to return and wine growers will indeed need to adapt to the new unpredictable and more extreme conditions.

The Effects on Pests and Diseases

The certainty and specificity with which the interviewees described the weather changes was replaced by more vague descriptions concerning pests in the vineyards. "We noticed some change but nothing worrying" (R4) and "new insects need to be monitored" (R3) do not allude to specific pests, although R6 specified that he had had red spider and other parasites since 2010, but as the pest load on a vine depends on the exact weather conditions and as these vary, then so does the pest load. The pests are not considered too much of a threat as "for now they are not invasive enough" (R1).

Diseases of the vine elicited more specific responses from the farmers. They mentioned Peronospora (downy mildew) (R2, R7) and esca (R6) as being particularly troublesome when weather is more humid than before, particularly when the African anticyclone replaces the previously common Azores anticyclone (R3). Their experiences of problems with mildew support the findings of work carried out with Italian farmers in 2015 [29].

3.2.2. The Adaptations Required When Grape Growing

Unpredictable weather as well as extreme weather events have led to the smallholders becoming much more reactive to their environments and proactive in their activities. This is demonstrated through later pruning to protect their "sleeping" (R1) vines from late frosts, leaving more grapes on the vine (R4) and harvesting a month earlier to avoid having an excessive sugar content in the grapes at harvest (R1), and taking measurements daily (R3) to determine when to carry out treatments and which products to apply (R2) rather than relying on the calendar as they did in the past (R3).

Certain changes have been made to the fabric of the vineyards themselves with cover crops being introduced to counteract the problem of soil erosion from waterbombs (R5) and straw being considered to improve the soil texture (R5).

Water was a strong theme in the interview data. Most of the farmers said they do not irrigate their land but have changed their treatment methods so that they now use less water than before when carrying phytosanitary treatments (R5, R6). Collecting water in dams was mentioned by R2 and R3 to provide "emergency water". There is a strong theme of trusting nature and the land to help mitigate the problems faced articulated by R1, who hopes that the vineyard's clay soil will retain water and the local spring will continue to provide for his vines' needs.

Others take a more pragmatic approach and explain that they have sought or are seeking help from professionals "with more expertise than us" (R7) as "we do not know

where to start" (R7). The sentiment that the "homemade" old way where "everything was decided by feeling" (R5) is not sufficient anymore and expert analysis and advice are now required in the face of the uncertainties due to climate change.

Regarding long term adaptations, none of the farmers mentioned considering growing different grape varieties, although R7 did suggest potentially changing from cultivating grapes to rice. Only R4, who currently farms in a privileged geographical position, mentioned the possibility of studying "new places to plant the vineyard". Rather than long-term strategies, their adaptations are based reactively on their day-to-day interactions with the vines and the land which they seem to inherently trust to find solutions; as R2 commented, "the vineyard is still following its course". Finance is a huge impediment to implementing change as the vintners "do not have infinite money" (R4).

Comparing these findings to the literature, it is clear that there is a partial divergence for the first time in the accounts provided. We noted an alignment with the findings of Mira de Orduna [37] and Ashenfelter and Storchmann [41], which stated that harvests were being brought forward in a bid to counter the rising sugar levels. Indeed, this was mentioned by all the interviewees, and was felt keenly at harvest time and is a marker in the calendar, so its advancement is a hugely visible sign of climate change. However, the divergence between the literature, in particular the strategies outlined by Mozell and Thach [14], and the interviewee data was quite marked and is presented in Table 1 below with the responses in bold which were mentioned by the farmers.

**Table 1.** Adaptations adopted in the vineyard.

| **For Vineyards** |
| --- |
| Change canopy management |
| Harvest at night |
| **Grow cover crops** |
| **Reduce water usage** |
| **Improve soil** |
| Consider deficit irrigation strategies |
| Consider light-abating cultural practices |
| Re-orient rows |
| **Increase vine crop load** |
| Use nitrogenous nutrients |
| Change variety |
| Train vines differently |
| Implement an integrated pest management system |
| **Move poleward or to a higher site** |
| Reduce carbon use |

Two of the adaptative strategies recommended by Mozell and Thach [14] which the Tuscan farmers implement already in the vineyards (see Table 1 above) concern water: one increasing water retention of the soil through growing cover crops and the other reducing water usage more generally. Farmers also mentioned improving their soil (which is a normal practice for organic farmers, which many of these are) and increasing vine crop load. Irrigation and building of dams to retain water were both mentioned but not specifically deficit irrigation systems as indicated above. Moving to a different location was mentioned by one farmer but for the others neither this potential adaptive strategy nor any of the others was mentioned in relation to combatting the effects of climate change in the vineyard. Instead, the farmers focused on their reactivity to the conditions—something which the literature did not allude to.

### 3.2.3. The Effects of Climate Change on Wine Making

Higher temperatures in the vineyard are producing an increase in volume of grapes with a higher pH (R1) and sugar content (all interviewees mentioned this, some repeatedly) despite the grapes being harvested earlier. This higher sugar content leads to wines with a fuller body and higher alcohol levels (R4). These findings correspond to Mira de Orduña's [37] conclusions, as does the higher temperature of the grapes arriving in the cellar at harvest time, which was a recurrent theme in the participants' accounts.

### 3.2.4. The Adaptations Required when Wine Making

In order to address the high alcohol levels and to produce "elegant" (R4) wines, the predominantly organic vintners are selecting yeast strains (R1, R3 and R4). The vintners are now facing "insanely high" (R2) temperatures in the cellar at harvest time and have already invested or are considering investing in cooling systems "to avoid possible temperature fluctuations" (R2).

This reactivity contrasts with ignorance, vagueness and passivity when looking at longer term investments in the winery. Comments such as "At the moment we do not have a real plan . . . we deal with the problems on a daily basis" (R5), "we do not have any big investments ready" (R1) and "we have to live with it" (R5) demonstrate an inertia fed by a lack of knowledge about the adaptations they could implement. The vintners expressed a sense of feeling destabilised and worried about the winemaking future as they seem to be facing so many unknowns and the practices which seemed immutable can no longer be relied on. However, this acknowledgement that "we will have to change many things" (R1) remains vague and framed in the future rather than the present. While some are seeking knowledge from "experts" (R7) who will be able to assist in the decision-making process and make their enterprises more financially stable to enable future investment, others remain much more passive and vague, as R1 remarked: "probably we will change our strategy". Again referring to Mozell and Thach [14]'s strategies (see Table 2 below), it is clear that half (those in bold) have been implemented by the vintners in this study.

**Table 2.** Adaptations adopted for wine production.

| **For Wine Production** |
| --- |
| **Use cooling equipment** |
| Inspect cellar hygiene |
| **Change to a more alcohol-tolerant yeast** |
| **Use sugar-reducing techniques** |
| **Use acidification** |
| Blend with wines from other terroirs and regions |
| Leave white wine of their lees longer |
| **Schedule harvest activities early** |
| Use renewable energy |
| Consider an energy consumption-monitoring plan |

As mentioned above, the winegrowers all harvest earlier and have all noticed that to reduce sugar they are changing yeast and using acidification. To combat the heat of the harvested berries, they use cooling techniques or are planning to. None of the interviewees however referred to measures related to energy nor did they mention blending wines. It is not known if cellar hygiene is considered to be an adaptative strategy as, again, it was not mentioned by our interviewees, nor was the recommendation related to white wine remaining on their lees.

A frequent comment was made by the farmers about the problematic major investment required for adaptations, backing up the findings of Poirier et al. [15] who noted that smaller

operations are less likely to be able to adapt than larger ones, which is especially appropriate for contexts such as this where most winemakers are small, family-run affairs.

*3.3. How Does Climate Change Affect the Agritourism Business for Tuscan Wine-Producing Agritourism Owners and How Do They Adapt to These Impacts?*

One positive aspect which the agritourism owners expressed relating to the impact of climate change on the tourism aspect of their enterprises was that warmer temperatures year-round mean that "now we are open all year" (R1), confirming the literature on the shoulder season [12,53–55]. However, this was nuanced by R6 who articulated that more generally "climate conditions could really damage the tourism in the area" and the unpredictable weather conditions in the area discourage people to book in spring and autumn (R2 and R3). This leads R3 to conclude that "it is more negative than positive". The hot temperatures in summer are not necessarily a godsend either, as expressed by R6 who sees his potential guests head for the sea, thus supporting the findings, specific to Tuscany, of Cai et al. [56]. R8′s potential solution to this problem by installing a swimming pool for his guests is in contradiction with Mozell and Thach's [14] recommendation to reduce water use and energy consumption. The use of air conditioning is mentioned as something which R8 reluctantly admits might become necessary instead of the existing ceiling fans echoing Perry's [73] findings regarding tourists' expectations. The participants' apparent lack of awareness of the unsustainable nature of these two adaptations is striking.

Wet weather led to late cancellations: "when it's raining [guests] start cancelling" (R6) supporting Cai et al. [56]'s work who indicate that this is less of a problem with the more "resilient" international tourists for whom it is more complicated perhaps to cancel at the last minute.

The marketing of agritourisms divided the participants. R5 seemed very attuned to the added value of organic and sustainable practices for his agritourism business in line with Embacher [74]; he described the tourists as loving his ancient pasta-making mill and ensured he marketed his installation of photovoltaic panels. Contrastingly, R3, who had also installed solar panels, said he had done so for himself and was "not very good at marketing this kind of thing". R5 demonstrated an understanding of some of his customers by commenting that "everyone is speaking about it . . . this can get some guests", while R7 noted that "in our area there are many arts cities and spectacular nature. I believe that tourism will exist also if the climate changes", demonstrating a reliance on the beauty and culture of the location and a passivity also noted regarding vineyard practices.

## 4. Conclusions

There is no doubt that climate change is real and is having an impact on grape growing, wine production and agritourism in Val d'Orcia. The small-scale wine producing entrepreneurs interviewed for this project all recognize that they need to change their practices in relation to their grape growing, wine production and agritourism as a result. Some of these actions are clear to them as they can be implemented from a reactive starting point, but the farmers are all floundering when looking long term at the adaptations they can implement in their enterprises, be they to do with the wine making or the agritourism side of the business. There is a burgeoning use of professional advice in the wine-producing sector, but experts are less considered with regard to the agritourism part of the business, and there is an absence in the interviews of mention of political leadership on this matter. There is also a passivity regarding marketing the organic and sustainable approaches despite them being appreciated by their guests. As the impacts of climate change are going to continue, it would seem prudent for these farmers who are so invested in the simple traditional life, working and respecting their family's land, to turn to professionals to help them invest wisely in adaptations in all of the areas of their businesses, and not only in the wine-making sector, which will enable them to continue doing what they love in a place of spectacular beauty. This could be in the form of a collective of agritourism owners so that best practices could be shared in a more economic way than by individuals each turning to

a different advisor, but there also needs to be recognition and action on behalf of politicians regarding the implications of the climate crisis to support these farmers into the future.

**Author Contributions:** Conceptualization, R.G.; Methodology, R.G.; Validation, R.G.; Formal analysis, R.G.; Data curation, N.M. All authors have read and agreed to the published version of the manuscript.

**Funding:** This research received no external funding.

**Institutional Review Board Statement:** The study was conducted in accordance with the Declaration of Helsinki, and approved by LES ROCHES CRANS-MONTANA as part of a bachelor's dissertation in 2021.

**Informed Consent Statement:** Informed consent was obtained from all subjects involved in the study.

**Data Availability Statement:** Please contact the first author for access to the data used in this study.

**Conflicts of Interest:** The authors declare no conflict of interest.

## Appendix A. English Interview Schedule

1. How has climate change modified your practice in the vineyard since you have been working in this business?
   a. Do you use a different amount of water?
   b. Do you use different pesticides?
   c. Did you notice different parasites?
   d. Did you witness the development of extreme weather events?
2. How has climate change modified the wine-making practice since you have been working in this business?
3. How has the quality and characteristics of your wine changed since working in this business?
   FOLLOW UP: Do you think any of these changes might be due to climate change?
4. Are you planning any changes in the winemaking process or in the work in the vineyard to counter the effects of climate change?
5. Which do you rely on more financially, agritouristic activity or selling wine? Could you do one without the other?
6. To what extent do you think the reputation of your wine influences the success of your agritourism?
7. Do you think the effect of climate change could positively or negatively affect the success of your agritourism? How?
8. Have you already made any changes to your business strategy as a response to climate change? How?
9. Are you planning any new business strategy as a response to climate change? How?
10. If the consequences of climate change were to continue or even get worse, how would it affect your business strategy?

## Appendix B. Italian Interview Schedule

1. In che modo il cambiamento climatico ha modificato il lavoro nella vigna da quando lei hai iniziato l'attività?
   a. Utilizza una diversa quantità d'acqua?
   b. Usa pesticidi differenti?
   c. Ha notato nuovi parassiti?
   d. Ha notato uno sviluppo di insolite ed estreme condizioni meteorologiche?
2. In che modo il cambiamento climatico ha modificato il lavoro in cantina da quando hai iniziato l'attività?
3. Come sono cambiate qualità e caratteristiche del suo vino da quando hai iniziato l'attività?

A SEGUIRE: Pensa che queste modificazioni possano derivare anche dal CC?

4. Sta pianificando modifiche nel lavoro in vigna o quello in cantina per contrastare i futuri effetti del cambiamento climatico?
5. Quale tra Agriturismo e Produzione di vino porta più introiti all'azienda? Potrebbe sostenere le spese di uno senza l'altro?
6. In che misura pensa che la reputazione del suo vino influenzi il successo dell' agriturismo?
7. Pensa che gli effetti del cambiamento climatico possano influenzare positivamente o negativamente il successo del suo agriturismo? Come?
8. Ha già modificato il qualche modo la tua strategia aziendale in risposta al cambiamento climatico?
9. Sta pianificando modifiche alla strategia aziendale in risposta al cambiamento climatico? SE SI: Quali
10. Se l'impatto del cambiamento climatico diventasse ancora più forte, in che direzione pensa andrà la sua azienda?

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
