# Peer review of "Perceiving and Adapting to Climate Change: Perspectives of Tuscan Wine-Producing Agritourism Owners"

_sustainability, doi:10.3390/su15032100_

Round 1

Reviewer 1 Report

Thank you for the paper, which I enjoyed reading greatly having visited the region recently. 

The review of the literature specific to wine tourism, and to climate issues in agritourism, viticulture and agriculture more generally, is current and well prepared.

The analysis of the excellent interviews is doen systematically using appropriate software and is reported very clearly and in a well structured way.

The one criticism I have is that the viticulture and enotourism elements are not well entwined, but rather there is a focus on the viticulture challenges, and some lesser discussion of the tourism issues, but these seem quite distinct. However, this is a minor concern.

A few (pednatic) suggestions:

p. 1, line 7, that instead of the; line 9 pruction instead of producing; line 19 formerly instead of formally

p. 2, line 74, adaptable instead of adapted

p 5, line 207 moment [is] really 

p 9 no need for the titles to be in the tables as well as above the tables - remove first line; line 374 have implemented instead of implement

Author Response

Thank you for your positive comments.

All but two of the 'pedantic' (but thank you) corrections have been made - the only ones which remain are adapted and implement as they are correct in the form they are in. Thank you for the other suggestions.

I have modified the introduction to the RQs (line 137) so that the distinct way they were addressed is not a surprise to the reader.

Thank you for understanding the nature of this article, I appreciate the feedback.

Reviewer 2 Report

The authors have documented the perceptions and adaptations of wine growers and agritourism. The initial presentation of the paper is quite appealing when considering segregated aspects of viticulture and agritourism, however, later the perceptions as well as adaptation to climate change for the two groups are not salienltly handled nor differentiated. The research questions raised are pertinent: 

How do Tuscan wine-producing agritourism owners perceive climate change?

How does climate change affect grape growing and wine-production for Tuscan wine-producing agritourism owners and how do they adapt to these impacts?

How does climate change affect the agritourism business for Tuscan wine-producing agritourism owners and how do they adapt to these impacts?

There should be a valid question as to why the climate change for wine growing is necessary as there are plenty of studies done to consider such aspects in agriculture across the globe - from micro to macro level, using field to historical data. The other aspect relates to this study's selection of sample where no randomness has been ensured to generalize the findings. Moreover, the findings are just simple assertions without any statistical validation nor examining the perceptions or adaptations dynamics among various groups and the frequency of both perceptions and adaptations as to how many of them perceive such issues and how they adapt. There are many studies which give comprehensive tabulation of perceptions vis-a-vis adapation. The Results section too is terse to portray reality while the conclusion section is unusually too lengthy to distract readers.

Author Response

Thank you for your review.

Regarding the literature, climate change is indicated as clearly a problem for agriculture in general and for wine growers and world wide and in Italy where found in 1.1 so we are unsure what needs to be done further here. The references are, as far as we can ascertain, logical, relevant, and where possible, up to date. We cannot identify any which aren't. Could further guidance be provided here if it is still felt that the literature is insufficient?

Regarding the comments related to the methodology, the article states that the research approach was a case study. As explained in Clark et al. (2021) 'the case study design does not aim to produce generalisable findings' (p. 61) unlike survey research.  The same response applies to statistical application to the data  - again, it is not appropriate for this sort of approach and would have been wrong to apply.

We were disappointed to read that you felt the results were 'terse' as an attempt was made to do justice to the respondents by letting their words sing through the text, using quotes to "support researcher claims, illustrate ideas, illuminate experience, evoke emotion" (Sandelowski, 1994).

Regarding the conclusion, it is challenging to determine what to remove as all three of the RQs are addressed in detail here. Could more specific information be provided?

Thank you.

Clark, T., Foster, L., Sloan, L., & Bryman, A. (2021). Bryman’s social research methods (6th ed.). Oxford University Press.

Sandelowski, M. (1994). The use of quotes in qualitative research. Research in Nursing & Health, 17(6), 479–482. https://doi.org/10.1002/nur.4770170611

Reviewer 3 Report

• What is the main question addressed by the research? It is now widely accepted the climate change is having a profound impact on the weather 7 systems around the world. These, in turn, have a considerable effect on two important elements of 8 the Tuscan economy: wine producing and tourism. This case study sought to explore the relationship between the perception of Tuscan wine-producing agritourism owners of the potentially abstract notion of climate change and their concrete experiences as entrepreneurs. Semi-structured 11 interviews were conducted with eight wine-producing agritourism owners or managers in Val 12 d’Orcia, a small area of Siena, Tuscany and analyzed thematically. The impact of climate change on 13 the area’s viticulture is undeniable but the responses to the challenges are more nuanced. Political 14 leadership on the climate crisis appears absent and perhaps as a consequence, these small-scale op-15 erators lack knowledge and funds to enable them to plan ahead: they react often day-to-day to the 16 immediate weather conditions rather than planning long term. While recognizing the difficulties 17 they face from climate change as viticulturists, as agrotourism owners they welcome the longer sea-18 sons which enable them to open in the formally barren shoulder seasons but struggle with last-19 minute cancellations due to unpredictable weather in the area. • Do you consider the topic original or relevant in the field? Does it address a specific gap in the field? The subject it deals with is original, but the authors do not manage to record the contribution of the article to the international literature. • What does it add to the subject area compared with other published material? The contribution of the article is not clear, the authors do not report it. • What specific improvements should the authors consider regarding the methodology? What further controls should be considered? The methodology part of this paper is absent, the authors do not analyze what methodology they follow. • Are the conclusions consistent with the evidence and arguments presented and do they address the main question posed? The authors should put more effort and thoroughly discuss point estimates, estimated effects, and the intuition behind the results backed up by the literature. • Are the references appropriate? No

Author Response

The comments below are addressed individually:

Do you consider the topic original or relevant in the field? Does it address a specific gap in the field? The subject it deals with is original, but the authors do not manage to record the contribution of the article to the international literature.

Literature from around the world was used in the article as well as specific to the country of the study. The importance of agritourism to the case study area was indicated. We are not sure what more we could have done here.

• What does it add to the subject area compared with other published material? The contribution of the article is not clear, the authors do not report it.

Lines 100-102 and 128-139 now address the gaps in the literature.

• What specific improvements should the authors consider regarding the methodology? What further controls should be considered? The methodology part of this paper is absent, the authors do not analyze what methodology they follow.

Section 4.4 has been added. The  methodology was addressed in 2. We stated how we gathered our data and how it was analysed in lilne with the case study approach. We say which software and which sort of analysis we used. We explore insider research. These are all appropriate to our study and are explained. Is there anything here which is unclear? 

• Are the conclusions consistent with the evidence and arguments presented and do they address the main question posed? The authors should put more effort and thoroughly discuss point estimates, estimated effects, and the intuition behind the results backed up by the literature.

Our article is based on interview data. There are no point estimates or no estimated effects in the data as we wanted to explore the way the respondents perceived climate change not the concrete scientific effects -which have been noted already in the literature. The way the data were reported is, we feel, appropriate for a qualitative, interview-based, approach. 

• Are the references appropriate? No

We do not understand this comment as the literature relates closely to the subject at hand. None of it is, as far as we are aware, unappropriate so this comment is a puzzle.

The article was written by a native English speaker so the comment that the text needs extensive editing is also a little perplexing. Some changes have been made where there were typographical errors.

Round 2

Reviewer 2 Report

The article still lacks a scientific vigor and fluency to advocate a convincing insights related to novelty as well as consistency with the previous work. A lot of work is required to support the hypothesis/background as well as identifying the research gaps and the methods used thereof. The discussion still lacks a flow to make succinct but meaningful ideas.

Author Response

I would like to thank you for your comments. I have, I think, attended to all of them and I hope the article is stronger as a result.

I thank you for your patience and wish you a delightful 2023.

Reviewer 3 Report

The authors have made very few changes in the current version of their article compared to the previous version.

Author Response

Thank you for your comments. The paper has been revised once again but I thank you for reconsidering your original comments and wish you a delightful 2023.

Round 3

Reviewer 2 Report

I am delighted to see a much improved version that is enough for consideration of publication by the journal.

Reviewer 3 Report

"The final version of the paper has improved. "